# Astrocyte morphology is confined by cortical functional boundaries in mammals ranging from mice to human

Raya Eilam[1], Rina Aharoni[2], Ruth Arnon[2], Rafael Malach[3]*

[1]Department of Veterinary Resources, The Weizmann Institute of Science, Rehovot, Israel; [2]Department of Immunology, The Weizmann Institute of Science, Rehovot, Israel; [3]Department of Neurobiology, The Weizmann Institute of Science, Rehovot, Israel

**Abstract** Cortical blood flow can be modulated by local activity across a range of species; from barrel-specific blood flow in the rodent somatosensory cortex to the human cortex, where BOLD-fMRI reveals numerous functional borders. However, it appears that the distribution of blood capillaries largely ignores these functional boundaries. Here we report that, by contrast, astrocytes, a major player in blood-flow control, show a striking morphological sensitivity to functional borders. Specifically, we show that astrocyte processes are structurally confined by barrel boundaries in the mouse, by the border of primary auditory cortex in the rat and by layers IIIa/b and Cytochrome Oxidase (CO)-blobs boundaries in the human primary visual cortex. Thus, astrocytes which are critical elements in neuro-hemodynamic coupling show a significant anatomical segregation along functional boundaries across different mammalian species. These results may open a new anatomical marker for delineating functional borders across species, including post-mortem human brains.

**\*For correspondence:** rafi.malach@gmail.com

## Introduction

The cerebral cortex has a remarkable capacity to regulate its own blood supply according to local neuronal activity demands (*Buxton and Frank, 1997*). This mechanism is of great interest, since it is the basis of optical imaging of intrinsic signals (*Vanzetta and Grinvald, 1999*) as well as BOLD-fMRI imaging (*Logothetis et al., 2001*). In particular, blood flow modulations have been used to demarcate functional columnar boundaries (*Blasdel and Salama, 1986*; *Grinvald et al., 1986*; *Lieke et al., 1989*). A particularly striking example of such modulation has been found across the boundaries of the rodent's vibrissae related 'barrels' (*Cox et al., 1993*; *Derdikman et al., 2003*; *Petersen and Sakmann, 2001*; *Woolsey et al., 1996*; *Woolsey and Van der Loos, 1970*; *Yang et al., 1997*). Given the tight coupling between columnar neuronal activation and blood flow, one could envision that the cortical vascular architecture should show anatomical shaping according to these boundaries. Such structural confinements by columnar boundaries were previously demonstrated in the dendritic and axonal arbors of barrel field neurons. These dendrites and axons appear to be 'repulsed' as they approach the barrel's boundary (*Brecht and Sakmann, 2002*; *Lendvai et al., 2000*; *Petersen and Sakmann, 2000*; *Shepherd et al., 2005*). However, detailed analysis of the micro-vascular organization in rodent barrel fields has failed to reveal such structural barrel-related modifications in the vascular capillary bed (*Blinder et al., 2013*). Similar failure of anatomical confinement was additionally reported in the primate visual cortex (*Adams et al., 2015*), including human cytochrome oxidase blobs evident in layers 2–3 of striate cortex, which constitute a robust example of functional 'mosaics' in the primate striate (*Livingstone and Hubel, 1984*). While the capillaries are a critical

**eLife digest** The brain is subdivided into many specialized regions that each has distinct roles. A key aim of brain research is to define the boundaries of these areas. Researchers have attempted to map the transitions between brain regions by identifying changes in the properties and activity of neurons (the cells that transmit information around the brain). However, these approaches cannot be used in some circumstances, such as when studying the living human brain, where only non-invasive experimental techniques can be used.

Cells other than neurons are also present in the brain. Astrocytes (a sub-type of glia cells) are support cells that have an extensive array of branches that project from each astrocyte's cell body, often giving it a characteristic star shape. Now, using high-magnification light microscopy, Eliam et al. show that the branches of individual astrocytes tend to avoid crossing the borders of brain regions with different roles. These changes in crossing densities define measurable boundaries between such subdivisions.

These density-change boundaries formed by the astrocytes are present in multiple species – mouse, rat and human – and in multiple systems: touch, auditory and visual. This discovery could provide a new window into the functional organization of the brain. It may also offer insights into how the brain optimizes its blood-flow control across different subregions.

The results of this study raise an additional question: is the confinement of astrocytes to single regions of the brain shaped by experience or is it present from birth? Exposing animals to different sensory experiences at different developmental stages will hopefully shed further light on this phenomenon.

component in controlling blood circulation, another major constituent that is hypothesized to play a crucial role in the coupling between neurons and blood flow are glial cells, particularly astrocytes. These cells provide a crucial bridge between neuronal activity and vascular blood circulation. The basic concept is that astrocyte branches are capable of sensing the level of neuronal synaptic communications, integrate this information and then transmit it to the blood vessels, with which they are in contact. It has previously been proposed that signals related to neuronal activity levels can control the capillary's diameter through astrocytes in this manner (*Attwell et al., 2010*; *Haydon and Carmignoto, 2006*; *Petzold and Murthy, 2011*; *Takano et al., 2006*). Importantly the role of astrocytes in blood flow control has recently been demonstrated using $Ca^2+$ imaging (*Otsu et al., 2015*). Interestingly, this research emphasized the importance of astrocytes' processes in blood flow control. Indeed, such control can be exquisitely precise, reaching sub-millimeter resolution (*Fukuda et al., 2005*).

Another interesting aspect of astrocyte morphology is the tight coupling between the neuronal and astrocyte processes. In particular, recent evidence has demonstrated the existence of tight anatomical co-localization of astrocyte processes and neuronal synapses in what has been termed a 'tripartite synapse' formation (*Araque et al., 1999*; *Halassa et al., 2007a*; *Perea et al., 2009*; *Santello et al., 2012*). Such links further suggest that astrocyte morphology may follow boundaries defined by neuronal functional compartments.

Finally, previous research has uncovered that in parallel to the neuro-vascular coupling- astrocytes play a significant role in activity dependent astrocyte-neuron lactate shuttle for the supply of metabolic substrates to neurons (*Pellerin et al., 1998*).

Given the crucial functional role of astrocytes in neuro-vascular coupling and other aspects of neuronal modulations, we hypothesized that the astrocytes, rather than the blood vessels, may show anatomical remodeling over time according to persistent functional boundaries. Indeed, in the case of olfactory bulb glomeruli were neuronal connectivity is highly ordered, dye coupling experiments highlight a preferential communication between astrocytes within glomeruli but not between astrocytes in adjacent glomeruli (*Roux et al., 2011*). To examine this hypothesis, we stained the entire astrocyte population in three species (mouse, rat and human) and in four functional boundaries (barrel field, auditory cortex, layer IIIa/b and Blob-interblob of human striate cortex), using an immunohistochemical procedure on thin paraffin sections. Superimposing the astrocytes onto the functional

boundaries revealed a significant structural relationship. Thus, astrocyte processes showed a significant anatomical confinement at the functional boundaries. This finding raises the possibility of using such astrocyte formations as an anatomical marker for the identification of functional boundaries across different cortical areas and species, including post-mortem human tissue.

## Results

### Reductions in the density of astrocyte processes are tightly localized to the barrel boundaries (septa)

Our examination of barrel field tissue was based on 154/146 (barrel/septa) measurements conducted in 4 sections in five left hemispheres of five mice (sectioned at 8 μM) and on 56/54 (barrel/septa) measurements calculatedin two additional left hemispheres (sectioned at 4 μM). For inspection of the details of astrocyte morphology, tangential (parallel to the cortical surface) sections were collected from all cortical layers. Prior to astrocyte staining, the sections were photographed using dark field illumination to highlight the barrel field borders. *Figure 1A* provides a low magnification (3.2x) example of the barrel boundaries obtained in one animal (dark field illumination, inverted contrast). The barrel boundaries are clearly evident as dark septa surrounding a lighter core.

Following the barrel boundary demarcation, sections were immunostained by GFAP antibody to detect the astrocyte morphology. This staining revealed the branching patterns of individual astrocyte processes (*Figure 1C*). An example of the branching of astrocyte processes is shown at intermediate (40x) and high (60x and 100x) magnifications in *Figure 1E,G and H* respectively. The spread of astrocyte processes was not uniform, but tended to be highly modulated, forming patches of dense and sparse regions. To reveal the precise details of such modulations, and their possible relationship to barrel borders, we manually drew the distribution of astrocyte processes at high magnification (see Materials and methods). The astrocyte process drawings for the sample cases are shown at low, (20x, *Figure 1D*) and intermediate, (40x, *Figure 1F*) magnifications. As can be clearly seen in *Figure 1C,E,G and H*, reductions in the density of astrocyte processes were not randomly distributed, but overlapped the barrel boundaries (e.g. see arrows-heads *Figure 1C–H* and dashed line in *1H*). A high magnification confocal image demonstrates the fine processes of astrocytes reaching the barrel border and abruptly stopping (red arrows *Figure 1H*).

### Is capillary distribution largely uniform across the barrel septa and core?

To compare the distributions of the astrocyte processes to that of the blood vessel capillary bed, the blood vessel distribution (traced by FITC dextran) was examined (see Materials and methods). *Figure 1B* depicts the distribution of blood vessels for the same field shown in *Figure 1C*. Unlike the clear boundary-related gaps that were evident in the astrocyte process distribution, blood capillaries appeared to ignore barrel boundaries and their tangential distribution was largely uniform across both the barrel septa and core.

The astrocyte confinement within barrel borders was a ubiquitous and robust phenomenon, and was observed in all animals studied. In all cases, there was a clear reduction in septa crossing by astrocyte branches.

### Quantitative measure of astrocyte branches and capillaries crossing the septa and the barrel cortex in layer IV

To obtain a quantitative measure of this phenomenon, the number of astrocyte process crossings was separately counted in the barrel core and septa (*Figure 2* illustrates the procedure). Barrels, with clearly demarcated borders, having a length of between 350–450 μm obtained from dark field imaging, were selected for analysis. Based on these photographs, straight lines (375 μm length) were superimposed on the center of the barrel border, aligned with the border's orientation. For comparison, lines of identical length and orientation were then placed at the center of the barrel cores (*Figure 2C*). These septa and core-related lines were then superimposed on the corresponding drawings of the astrocyte processes (see illustration in *Figure 2A*) and the number of astrocyte branch crossings of these lines was calculated. The same procedure was repeated for blood vessel crossings (*Figure 2B*).

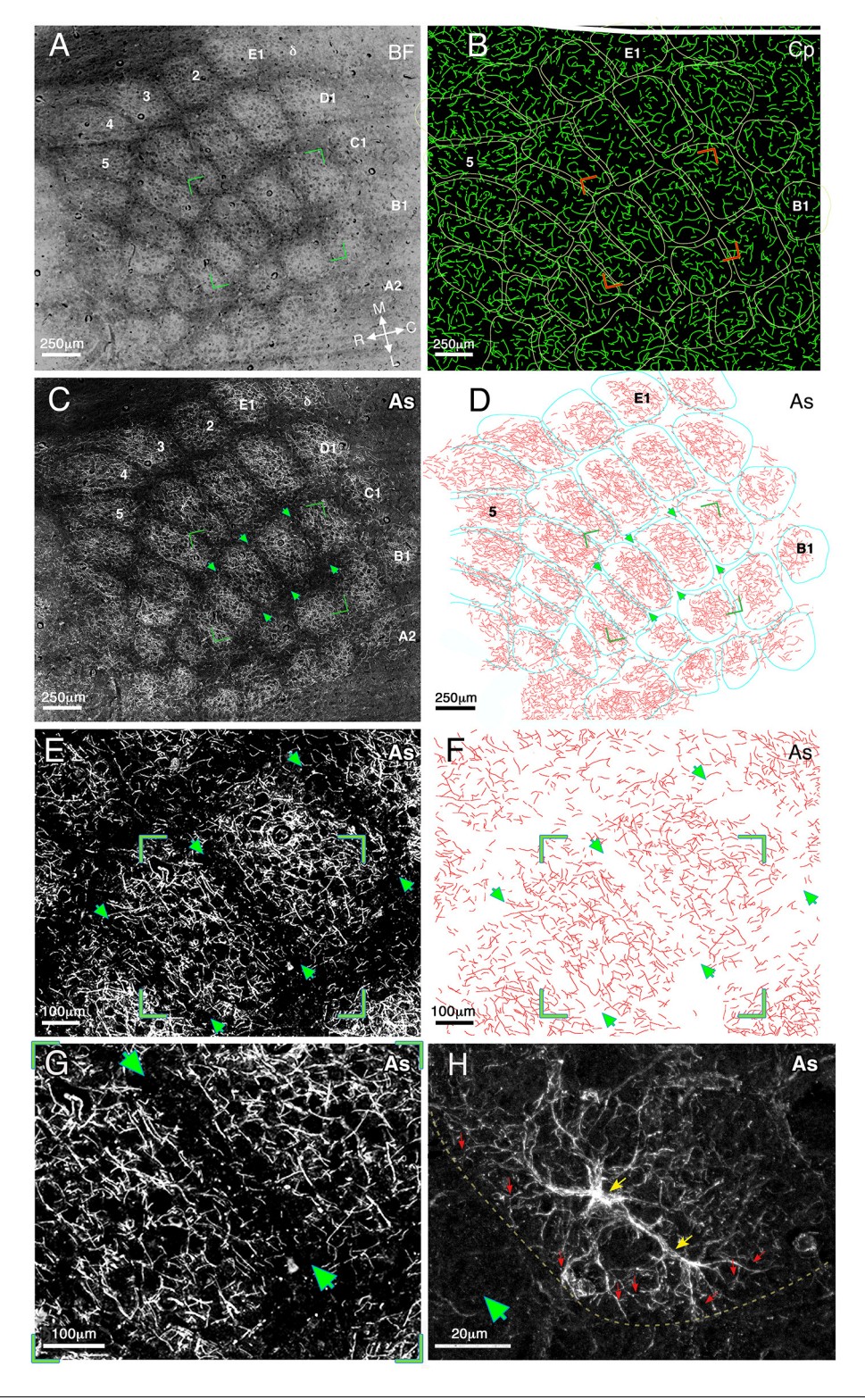

**Figure 1.** Relationship of astrocyte branches and capillaries to the barrels borders in cortical layer IV at medium and high magnifications. (**A**) A dark field illumination image (inverted contrast) of the barrel field (BF) area of a flattened left hemisphere. The barrel borders are evident as dark septa surrounding a lighter core. Barrel rows **A–E** and arcs 1–5 as well as the medial–lateral and rostral–caudal axes are indicated. (**B**) Reconstruction of the capillaries (Cp) visualized using fluorescein dextran. Barrel borders (derived from **A**) marked by yellow lines, were

*Figure 1 continued on next page*

*Figure 1 continued*

superimposed on the drawing of capillaries. The two images were aligned using penetrating blood vessels as landmarks. (C) Fluorescent micrograph of GFAP immunostaining (CY3 conjugated) of astrocyte processes (As). The image consists of 90 tiles that were assembled together revealing the lateral distribution of the processes. (D) A reconstruction of the astrocyte processes of image C performed by manual drawing of their branches at high magnification. The borders of the barrel field (marked in turquoise) and astrocyte processes were aligned as in B. Green arrowheads indicate three barrel borders shown in enlarged view in E and F. All the images were taken from the same tissue section. (E) Enlarged florescent image of stained astrocyte processes (same as in C) showing further details of the processes. Arrow heads point to a clearly visible gap in the astrocyte distribution at three barrel boundaries. Green corners indicate the enlarged image shown in G. (F) A reconstruction of the astrocyte processes (drawn manually at high magnification) shown in E. (G) An enlarged fluorescent image view of the area bounded by the green box in E, showing the astrocyte distribution relative to the barrel borders (arrows). (H) A high magnification confocal image demonstrates the fine processes of two astrocytes at the border between barrels and septa (green arrowhead) indicated by a yellow line. The astrocyte cell bodies are indicated by yellow arrows and examples of their processes by red ones.

The results of this analysis for 5 animals were performed on 4 neighboring sections with interval of about 20 μm between them. The number of astrocyte process crossings in $154 \pm 7$ barrel cores and $146 \pm 6$ septa, are shown in *Figure 2F*. There was a three-fold ($11.02 \pm 0.58$ *vs* $3.78 \pm 0.35$, means ± s.e.m, $t_{298} = 9.7$, p<0.001. 95% confidence interval, 10.26 to 11.82 and 3.3 to 4.2 respectively; Cohen's d = 1.72; *Figure 2F*, left) difference in the number of astrocyte crossings of barrel borders compared to the crossing numbers within the barrel's core. In contrast, there was only a slight trend of reduced crossings of blood vessels at barrel's border ($2.64 \pm 0.18$ *vs* $2.97 \pm 0.18$. 95% confidence intervals, 2.47 to 2.81 and 2.77 to 3.17) which did not reach significance ($t_{156} = 1.28$, p=0.2; *Figure 2F*, right).

## Quantitative measurements of astrocyte processes and capillary crosses in the septa and the barrel cortex in layers III and V

It should be noted that the barrel boundaries are less anatomically defined in the supra and infra-granular layers (*Durham and Woolsey, 1977*; *Woolsey and Van der Loos, 1970*). Consequently, our barrel boundary for the layers above and below layer IV was based on a vertical extrapolation of layer IV boundaries (relying on penetrating arterioles to align the borders. *Figure 2D and E* are representative drawings taken from layer III (D) and layer V (E). Although the distribution of astrocyte processes was suggestive of barrel confinement, the relationship to the barrel boundaries was much less evident.

Quantitative comparison of the astrocyte crossings within and across barrels of three animals (85 barrel cores and 76 septa), performed in a similar manner to that conducted for the layer IV analysis, showed a lower, albeit significant reduction in septa-barrel crossings, relative to crossings of the barrel core in supra-granular layer III ($16.2 \pm 0.8$ *vs* $11.2 \pm 0.7$, $t_{159} = 3.6$, p<0.001. 95% confidence interval, 15.6 to 16.8 and 10.7 to 11.7 respectively, Cohen's d = 1.84; *Figure 2F*, left). A similarly significant confinement was found in the infra-granular layer V; $9.4 \pm 0.8$ *vs* $14.3 \pm 0.7$, $t_{154} = 3.9$ p<0.001. 95% confidence interval, 9.0 to 9.8 and 13.7 to 14.9 respectively; Cohen's d = 1.92; *Figure 2F*, left).

In contrast, there were no significant differences in crossings of capillaries at the barrel's border when compared to the barrel core in either layer III ($1.66 \pm 0.25$ *vs* $1.69 \pm 0.18$; $t_{155} = 1.31$, p = 0.19. 95% confidence interval, 1.49 to 1.86 and 1.57 to 1.81 respectively, *Figure 2F*, right) or in layer V ($2.46 \pm 0.29$ *vs* $1.73 \pm 0.28$; $t_{153} = 0.29$, p = 0.77. 95% confidence interval, 2.28 to 2.64 and 1.64 to 1.82 respectively; *Figure 2F*, right).

Comparison of the ratio between crossings within the septa and barrel core between layers revealed that astrocytes in layer IV showed a significantly higher level of astrocyte boundary confinement when compared to the supra and infra granular layers ($0.79 \pm 0.07$ in layer III, $0.36 \pm 0.03$ in layer IV, and $0.68 \pm 0.05$, in V, using a one-way ANOVA on log-transformed ratios, followed by a Tukey post-hoc test ($F_{2,105} = 20.6$, p<0.001; *Figure 2G*, left). Conversely, comparison of the septa/barrel ratio of capillary crossings between layers, revealed nonsignificant changes ($0.91 \pm 0.2$ in layer III, $1.05 \pm 0.09$ in Layer IV, and $1.68 \pm 0.39$ and in Layer V) using the same statistical analysis as in 2G,

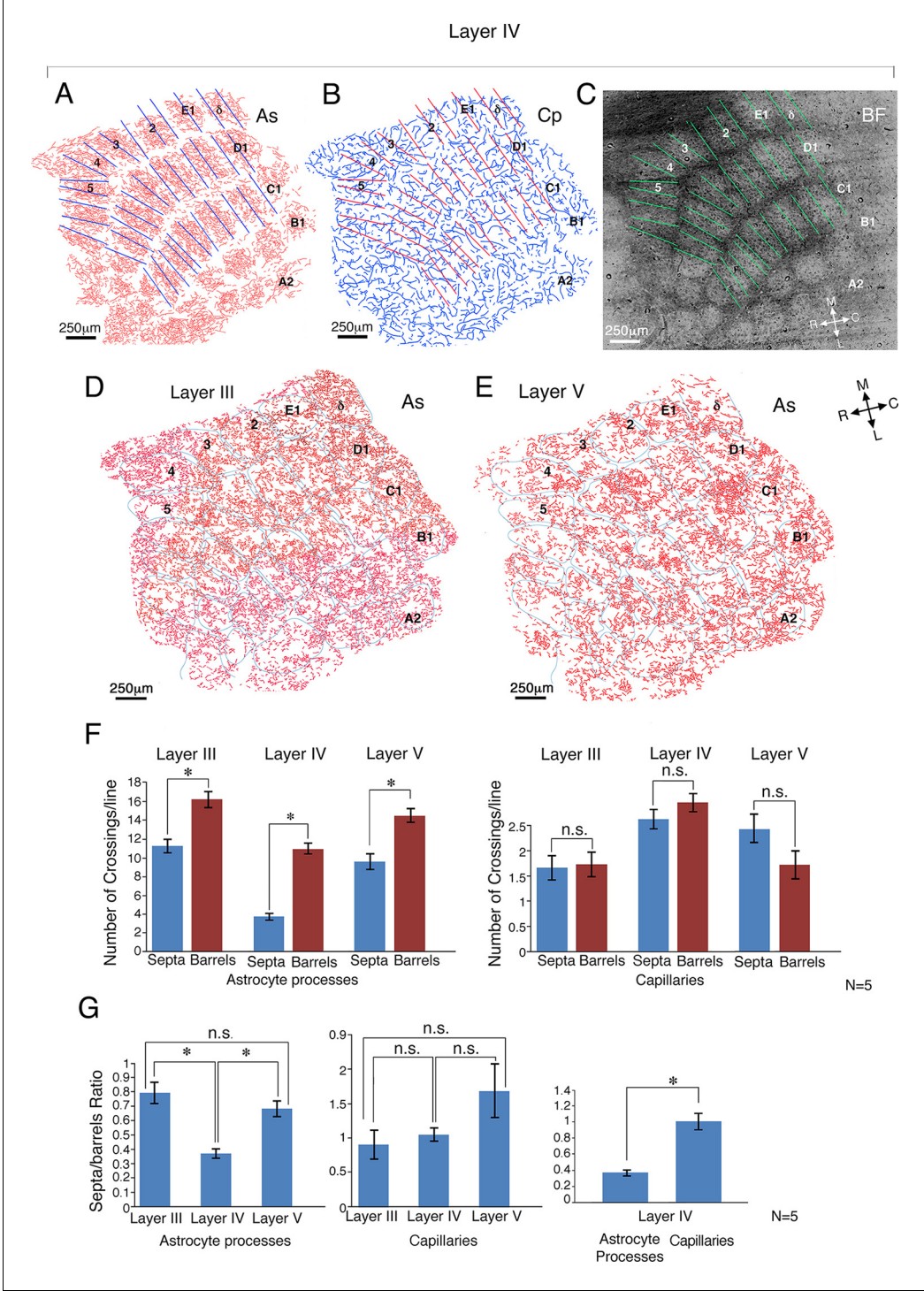

**Figure 2.** Quantitative comparison of astrocyte and capillary crossings in the core and septa of the barrels in layer IV as well as of the corresponding regions in the supra-granular and infra-granular layers. (**A**) Astrocyte process (As) reconstructions from the barrel field (BF) of layer IV. (**B**) Capillary (Cp) reconstruction of the same field. (**C**) A matrix of lines (375 μm length) was drawn at the center of the septa aligned with the border's orientation. For comparison, lines of identical length and orientation were superimposed at the center of the barrel cores (see Materials and methods for details). Septa and core related lines superimposed on the astrocyte and the capillary drawings in A and B respectively were aligned using penetrating arterioles. (**D** and **E**) Drawings of astrocyte distribution in layers III (supra-granular) (left) and layer V (infra-granular) (right) of the cortex, 80 μm above and below the barrel field borders. The boundaries of the barrels were transferred from the dark field image of layer IV

*Figure 2 continued on next page*

*Figure 2 continued*

and superimposed upon the astrocyte processes of layer III and V. All the images were aligned according to penetrating arterioles. The matrix of lines was also transferred from layer IV (not shown). (F) Histograms depicting the number of astrocyte processes and capillaries respectively, crossing the septal and barrel cores at layers III-V. (G) Histograms displaying septa/barrel crossing ratio (see Materials and methods) for the astrocyte (left) and capillary (middle) crossings, across the different layers, as well as a histogram comparing the ratio of astrocytes with capillaries in layer IV (right). Means ± s.e.m., *p<0.0001; n.s., non-significant.

left (*Figure 2G*, middle). Comparing the ratio between crossings of the septa and the barrel core of the astrocyte processes and capillaries in layer IV, the former was significantly lower ($0.36 \pm 0.03$ *vs* $1.05 \pm 0.09$, $t_{282} = 6.6$, p<0.001; *Figure 2G*, right).

To examine possible effects related to section's thickness, we compared 4 and 8 μm thick sections in layer IV). The analysis revealed the expected reduction of about 35% in the number of astrocyte process crossing in the 4 compared to the 8 μm thick sections, but the ratio between the septa/barrels for the astrocyte branches was similar in both thick and thin sections ($0.37 \pm 0.05$ and $0.36 \pm 0.03$ for 4 and 8 μm sections, respectively).

## Reductions in the density of astrocyte processes at the border of primary auditory cortex of the rat

While our results demonstrate a clear confinement of astrocyte processes to the barrel borders in the mouse cortex, it could be argued that such a phenomenon is unique to the barrel cortex or to the mouse. To examine how general this phenomenon is, we examined the density of astrocyte processes in the auditory cortex of the rat. The analysis was based on $82 \pm 9$ measurements on 4 neighbored sections (the interval between sections was about 20 μm) conducted in the left hemispheres of 4 rats (sectioned at Layer IV). For convenience, we will refer to all the secondary fields around A1 (*Polley et al., 2007*) as A2. The border of A1/A2 was defined by dark field illumination, the results of which are depicted in *Figure 3*. To examine the density of astrocyte processes in the A1 vs. A2 cortex, we obtained ten bands 25 μm wide and 100 μm long, their centers located at the borders between A1/A2 (*Figure 3C*, middle). Quantitative measurement of their density was taken along each pixel of the bands (*Figure 3C* right). In addition, we plotted straight lines (50 μm length), which were superimposed at a distance of 50 and 200 μm from the border of A1/A2 on both sides, aligned with the border's orientation. These A1 and A2 lines were then superimposed on the corresponding drawings of the astrocyte processes (*Figure 3E and G*) and the number of astrocyte branch crossings were calculated.

As can be seen from the histogram in *Figure 3C* right, there was a sharp and significant drop ($37 \pm 3\%$) in the density of astrocyte processes between A1 and A2. The histogram in *Figure 3C* left, reveals reduction in the number of astrocyte crossings immediately outside the A1 border (50 μm); $14.1 \pm 0.93$ *vs* $7.4 \pm 0.69$ ($t_{52} = 6.46$, p<0.001). Similar results were obtained when we compared between the two areas at a distance of 200 μm from the border; $13.96 \pm 0.54$ *vs* $7.18 \pm 96$ ($t_{50} = 5.86$, p<0.001). Importantly, no significant changes were observed at different distances from the border on either side, ($14.1 \pm 0.93$ *vs* $13.9 \pm 0.54$, $t_{52} = 0.25$, p=0.79, at a distance of 50 and 200 μm respectively in A1 and $7.48 \pm 0.69$ vs $7.18 \pm 0.45$, $t_{50} = 0.45$, p=0.64, 50 and 200 μm respectively in A2), indicating that the change in the density of astrocyte processes was abrupt at the A1/A2 border and did not gradually change approaching this border.

## Reductions in the density of astrocyte processes at the border of sub-layers IIIa / IIIb of the human V1 cortex

If the confinement of astrocyte processes at functional borders is a general phenomenon, such a finding could be particularly informative for studying the human cortex. Thus, examining the density of astrocyte processes may offer a novel 'window' into functional boundaries that may be revealed even in post-mortem tissue. To explore the possible extension of our hypothesis to the human cortex, we examined the boundaries of sub-laminae IIIa/IIIb of human primary visual cortex V1. Previous research has uncovered functional distinctions between these two sub-layers in the primate cortex (*Sceniak et al., 2001*). We analyzed coronal sections from the right hemisphere of one human post-

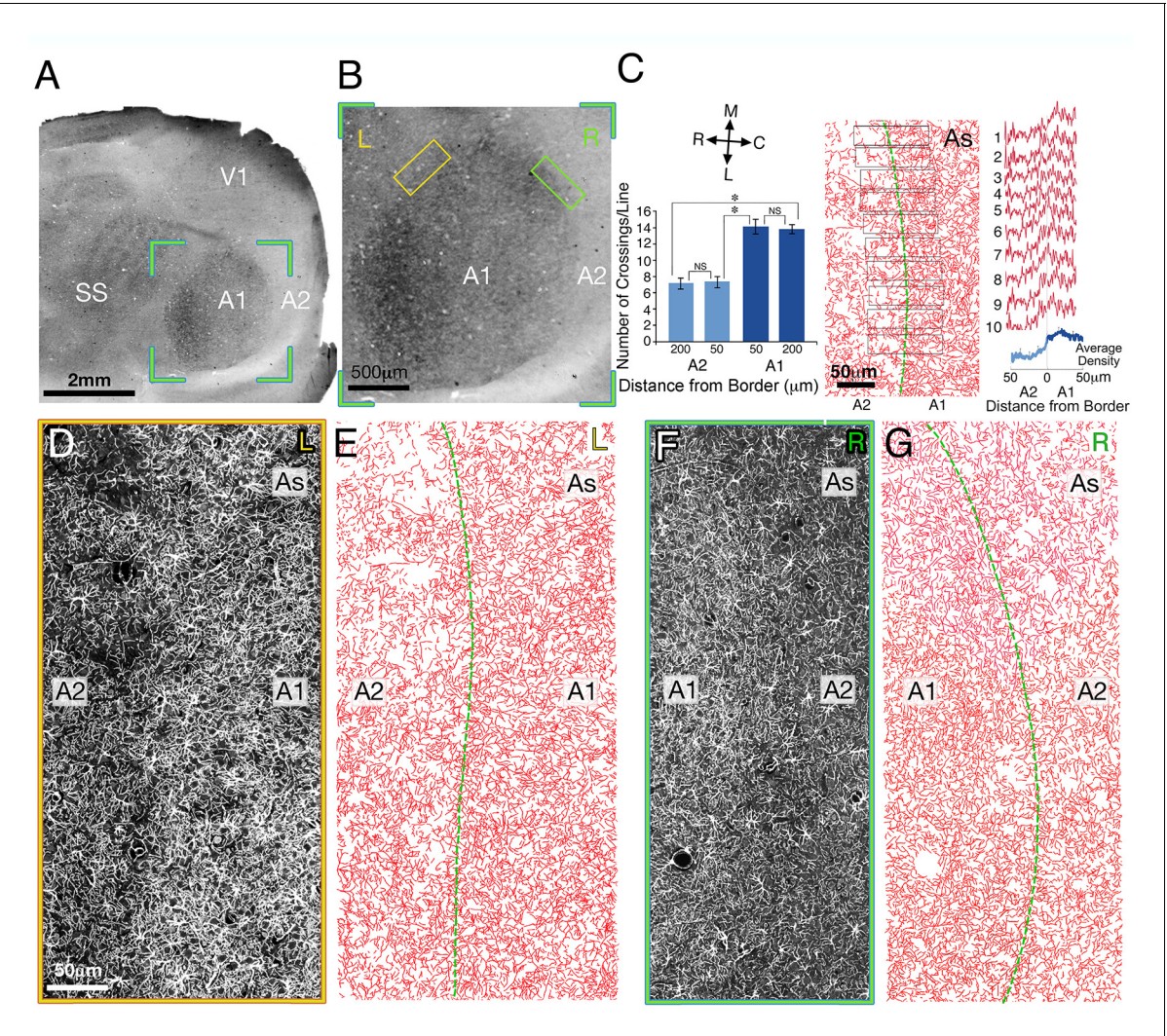

**Figure 3.** The relationship between astrocyte processes and the A1/A2 border in layer IV of rat auditory cortex. (**A**) Low magnification of a dark field illuminated image of a tangential section through layer IV. (**B**) Dark field illuminated image of areas A1 (primary auditory cortex, dark round area) and A2 (secondary auditory fields, bright area surrounding A1), taken from the green bounded region in **A**. (**C**) Left panel: graphical representation of the number of astrocyte processes (AS) crossing a matrix of lines (50 µm length) which were drawn at A1 and A2 areas, 50 and 200 µm from the A1/A2 border aligned with the border's orientation (not shown). Right panel: density plot measurements of ten 25 µm wide bands located perpendicular to the border's orientation (middle panel) in areas A1/A2 (individual traces-red, average-blue). (**D** and **F**) An enlarged fluorescent images of the astrocyte processes in the area bounded by the yellow and green boxes in **B** left (**L**) and right (**R**) border of A1/A2, respectively. (**E** and **G**) A manual reconstruction of the astrocyte processes in images **D** and **F**, respectively. The border between A1 and A2 is demarcated by a green line. SS, somatosensory cortex.V1, primary visual cortex

mortem brain (age 35). *Figure 4—figure supplement 1* depicts the orientation of the sections relative to the occipital pole. To reveal the IIIa/IIIb border we used non-phosphorylated neurofilament (NPNF) immunostaining (*Campbell and Morrison, 1989*; *Preuss et al., 1999*) and related this border to inter-laminar astrocyte processes (see *Figure 4D*). Astrocyte processes were not uniformly spread in layer III, but appeared to be constrained by the border of IIIa/IIIb. To quantitatively examine this effect, we manually drew the distribution of astrocyte processes at high magnification (*Figure 4C*). Measuring the density of astrocyte processes between the two sublayers revealed a dramatic and sharp reduction in the density (66 ± 4%) at the border of layer IIIb (*Figure 4E*, left). The number of astrocyte branches crossing straight lines (375 µm length, see Materials and methods) at both sides of the border was also calculated in 4 neighboring sections

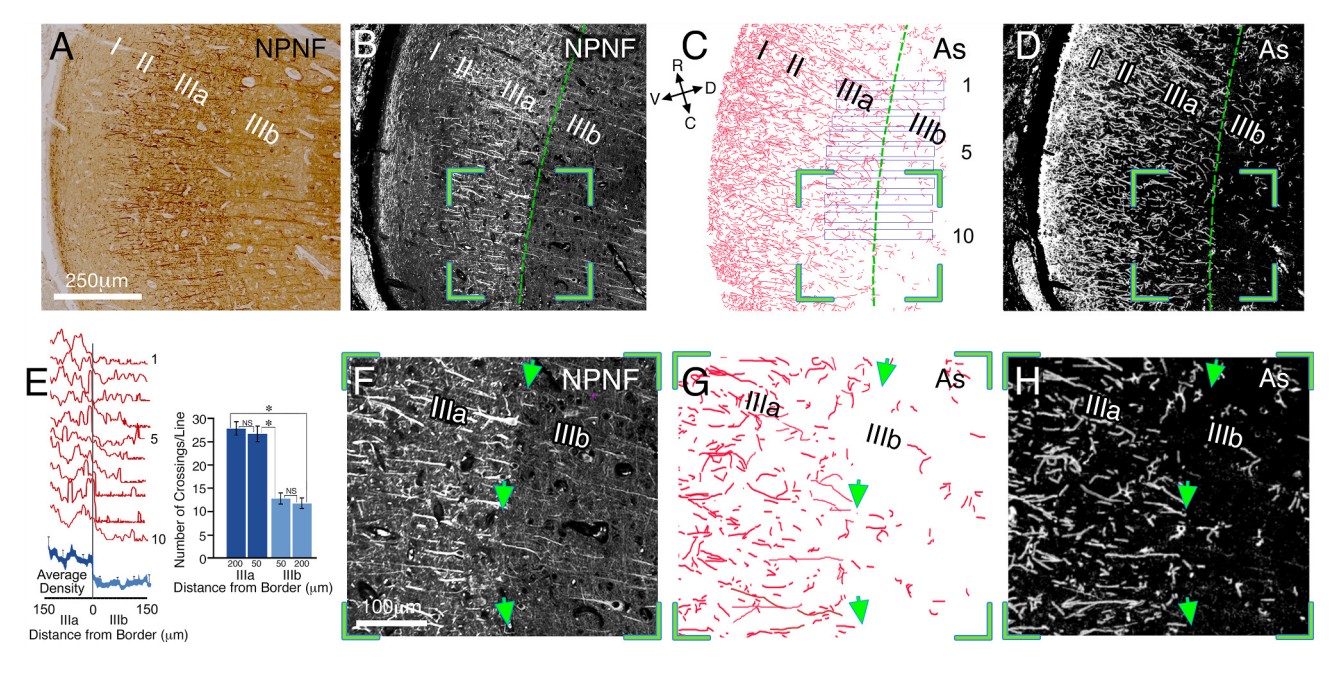

**Figure 4.** Relationship between interlaminar astrocyte processes and the sublaminar border of layers IIIa/IIIb in the post-mortem human striate cortex. Immunohistochemical staining of coronal sections of human area V1, illustrating the relationship between interlaminar astrocyte processes and the sublaminar border of layers IIIa/IIIb. (A) Bright field photograph of coronal section through V1 stained for NPNF (non-phosphorylated neurofilament), a higher magnification, image of the area is shown in *Figure 4—figure supplement 1D*. The border between IIIa/IIIb is demarcated by the heavily (II & IIIa) and lightly (IIIb) stained bands. (B and D) Immunofluorescent double staining for NPNF (B) and GFAP (D), respectively, on a section adjacent to that of A. The border between IIIa/IIIb is demarcated by a green line. (C) A manual reconstruction of the astrocyte processes (As) in image D. (E) Left panel: superimposed density plot measurements of ten 30 μm wide bands perpendicular to the border (plotted in C) in layers IIIa/IIIb (individual traces-red, average-blue). Right panel: a graphic representation of the number of astrocyte processes crossing a matrix of lines (375 μm length), which were drawn at layers IIIa and IIIb, 50 and 200 μm from the IIIa/IIIb border aligned with the border's orientation (not shown). (F–H) Enlarged image of stained pyramidal cell bodies and apical dendrites, as well as astrocyte processes taken from the green bounded region in B–D, respectively, showing further details of the astrocyte processes. Arrow heads point to a clearly visible IIIa/IIIb border

The following figure supplement is available for figure 4:

**Figure supplement 1.** Low magnification view of the human occipital lobe.

(interval of about 20 μm between sections, *Figure 4E*, right). There were no differences between the crossings of astrocyte processes in the same sub laminar layer at different distances (50 and 200 μm) from the border; $27.2 \pm 1.15$ vs $28.5 \pm 1.07$ for IIIa ($t_{68} = 1.66$, $p=0.10$) and $13.3 \pm 0.54$ vs $12.4 \pm 1.37$ for IIIb ($t_{62} = 1.18$, $p=0.24$). However, a highly significant change was found in the number of crossings when comparing the two sub-laminar areas at similar distances from the borders; $27.2 \pm 1.15$ vs $13.3 \pm 0.54$, $t_{66} = 22.39$, $p<0.001$, 50 μm, and $28.5 \pm 1.07$ vs $12.4 \pm 1.37$, $t_{68} = 18.44$, $p<0.001$, 200 μm from the border, indicating an abrupt change in astrocyte crossing at the sublaminar border.

## Reductions in the density of astrocyte processes in CO blobs of human striate cortex

Finally, we examined the potential segregation of astrocyte processes along the compartmental division of CO-blobs vs inter-blobs in human V1. It should be noted that, in contrast with the previous examples, the blob boundaries are not sharply defined and, in fact, dendritic trees, unlike barrel-field borders, are not 'repelled' by these fuzzy borders (*Malach, 1992*; *1994*). However, an interesting issue is whether the density of astrocytes significantly changes as one moves from the CO-blob into the inter-blob regions. To examine this question, we mapped neighboring sections (about 100 μm apart) of human post-mortem tissue with both the CO-blob locations, defined through CO

histochemistry (15 µm thick sections) and the astrocyte distribution through immunohistochemistry for GFAP. Images were aligned by matching blood vessels as landmarks.

Figure 5 depicts the results of this analysis. We counted the number of astrocyte crossings across each of the vertical 500 µm lines that were crossing blobs or inter-blobs area at layers II and IIIa (Figure 5D). As can be seen from the histogram (Figure 5E), there were no significant differences between the crossings of astrocyte processes in the inter-blobs at different distance from the blobs border; 11.2 ± 1.08 vs 12.1 ± 1.19 (p=0.58). However, we found a significant increase in astrocyte processes crossing at inter-blobs compared to the CO blobs proper (11.25 ± 1.08 vs 5.25 ± 0.72; t (48) = 9.1 p<0.001. 95% confidence interval, 10.33 to 12.17 and 4.57 to 5.93 respectively and 12.1 ± 1.19 vs 5.25 ± 0.72; t(48) = 11.2 p<0.001). 95% confidence interval, 11.28 to 12.92 and 4.57 to 5.93 (Cohen's d = 1.2 and 0.9 respectively. Figure 5E). Thus, we found a significant modulation in astrocyte density that follows the changes of CO-organization.

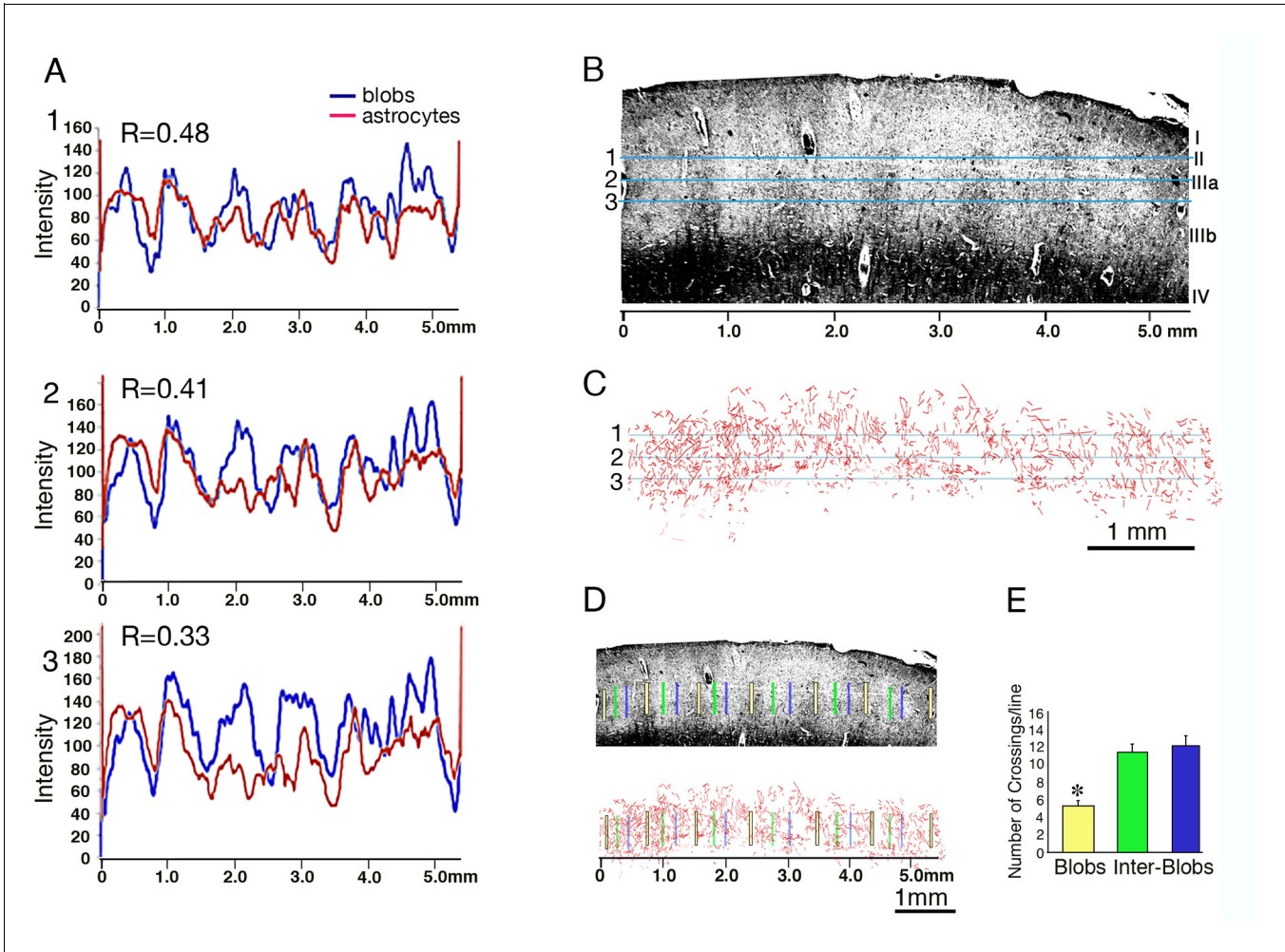

Figure 5. Relationship between CO-blobs and astrocyte density in the post-mortem human striate cortex. The relationship between the intensity of cytochrome oxidase (B, blue) and astrocytic GFAP staining (C, red) along cortical layer II and IIIa is depicted in a coronal section from human striate cortex. (A) 1–3; transmission plots of bright-field CO histochemistry (high-CO indicated by low levels of transmission, blue lines) and astrocyte process density (red). R-values (Pearson coefficients of correlation) indicate a significant CO-modulation of astrocyte density and blob-interblob organization. (B) A section stained for cytochrome oxidase shows a dense band in layer IV and periodic vertical regions of enhanced enzyme activity in the upper layers (blobs), interspersed with inter-blob regions at about 1 mm period. Horizontal lines (numbered 1–3) depict the density measurement lines shown in A1-3 respectively. (C) Astrocyte process reconstruction in layer II and IIIa of an adjacent section stained for GFAP. Horizontal blue lines, similar to B. (D) Same sections as in B and C, indicating the location of 500 micron vertical lines that were placed at the center of blobs (yellow), the center of inter-blobs (green) and at the left margin of inter-blobs (blue). (E) Histograms depict the number of astrocyte crossings across each of the vertical lines shown in D. Note the significant increase in astrocyte processes crossing in inter-blobs compared to the CO blobs proper (*p<0.005).

## Discussion

Our results reveal a novel structural manifestation of cortical architecture, reflected in the tendency of astrocyte processes to be anatomically confined by functional borders. The effect was significant and was found across different cortical areas and species, including human post-mortem tissue.

### Astrocyte processes are anatomically confined to functional boundaries: mouse barrel fields

First, we demonstrated a consistent anatomical confinement of astrocyte processes to somato-sensory barrel boundaries. In contrast to astrocyte confinement, and in agreement with previous studies (*Blinder et al., 2013*; *Woolsey et al., 1996*; *Wu et al., 2014*), the distribution of blood capillaries did not show a significant relation to barrel boundaries. In that sense, the astrocyte processes appear to be more similar to dendritic arbors of barrel neurons, which were also reported to show a clear anatomical restriction in relation to barrel borders (*Brecht and Sakmann, 2002*; *Feldmeyer et al., 2002*; *Lefort et al., 2009*; *Lendvai et al., 2000*; *Lubke et al., 2000*; *Petersen and Sakmann, 2000*; *Shepherd et al., 2005*).

In contrast to layer IV, where barrel boundaries were clearly demarcated, the columnar segregation in supra- and infra-granular layers was less distinct. This is reflected both in structural segregation and in neuronal signaling (*Feldmeyer et al., 2002*; *Lubke et al., 2000*; *Petersen and Sakmann, 2001*). It is interesting to note that such boundary 'blurring' was also reflected in the astrocyte geometry, since our quantitative analysis revealed a significant reduction in the anatomical confinement by barrel boundary, both in supra- as well as infra-granular layers (*Figure 2*). Importantly, a significant, albeit smaller decrease in border crossing was observed in these layers as well. This result suggests that the level of anatomical border confinement could be a gradual process rather than an all or nothing categorical 'toggle' of a columnar confinement process.

However, a word of caution should be noted here regarding the methodological accuracy of barrel border definition. In layer IV, such demarcation was straightforward, since barrel boundaries were easily demarcated in this layer (*Welker and Woolsey, 1974*; *Woolsey and Van der Loos, 1970*). Thus, determining the septa and core boundaries could be achieved with great accuracy. By contrast, defining barrel borders outside layer IV was less accurate, relying on vertical extrapolations from layer IV borders by alignment of some of the penetrating arterioles. Such extrapolation may have introduced some inaccuracies in our border delineations. Therefore, the estimates of border crossings in the upper and lower layers should be considered less accurate compared to our layer IV results.

### Astrocyte processes anatomically confined to functional boundaries: rat and human cortex

Our study reveals a significant dissociation between capillary distribution, which appears to ignore columnar boundaries, and the anatomical shaping of astrocyte processes. However, it could be argued that this observation was unique to the mouse barrel fields which indeed show a particularly striking functional segregation. To examine whether the astrocyte structural confinement may reflect a more general phenomenon, we extended our analysis to additional species (rat and human) and to additional functional boundaries (A1/A2 sub-layer IIIa/IIIb and blob/inter-blob organization respectively). In all cases (with the exception of CO-blob boundaries, which are not precisely defined to begin with, see (*Malach, 1992*), we found a significant and abrupt reduction in astrocyte processes near the regional boundaries (see *Figures 3–5*).

It could be argued that the staining of astrocyte processes was not complete although our procedure (thin sections, antigen retrieval, long time incubations) enhanced the staining sensitivity. While we cannot rule out this possibility, such limited staining could not account for the selective drop in the density of astrocyte processes that was observed near functional boundaries, and not away from them as seen in all three cases studied (see histograms in *Figures 2F*, *3C* and *4E*).

Thus, we can safely conclude that astrocyte process morphology can be confined by regional and columnar boundaries across species and systems. Of course, our limited sample does not prove that all cortical boundaries should have a similar impact on astrocyte structure, but the observed phenomenon opens a potentially informative direction of future research, in which such astrocyte boundaries could be mapped in great detail across species and brain areas.

It should be noted that a number of previous studies have demonstrated that astrocyte activity may be modulated according to the functional properties of the modules, within which they reside. For example, (*Schipke et al., 2008*) have demonstrated barrel specific $Ca^{2+}$ response in astrocytes. However, it is important to emphasize that such functional distinctions, while offering a possible source for the morphological restructuring reported here, do not on their own constitute a demonstration of such anatomical reshaping. For example, in the Schipke et.al. experiment, the astrocyte selectivity was rapidly abolished by $GABA_A$ receptor antagonist application, while the astrocyte process' reshaping clearly operates (if at all) on much slower time scales.

Similarly, previous studies have demonstrated synaptic and molecular selectivity of astrocytes- for example Houades et al. (*Houades et al., 2008*) have shown that gap junction density may be modulated along barrel boundaries, while Voutsinos-Porche et al. (*Voutsinos-Porche et al., 2003*) have shown a transient selective appearance of glutamate transporter in barrels which disappeared in the adult cortex. These prior studies further confirm the potential functional selectivity of astrocytes. However they are not relevant to the issue of the astrocyte morphology. Thus, selective aggregation of gap junctions may occur with or without anatomical confinement of astrocyte processes reported here.

Astrocytes have been suggested to play a critical role in neurovascular coupling, controlling the blood flow upon an increase in neuronal activity (*Attwell et al., 2010*; *Haydon and Carmignoto, 2006*; *Petzold and Murthy, 2011*; *Takano et al., 2006*).

Furthermore, Astrocyte processes have been demonstrated to manifest tight anatomical formations with neuronal dendrites and the activity of the brain actually arises from the coordinated activity of a network comprises of both neurons and astrocytes (*Araque et al., 1999*; *Halassa et al., 2007a*; *Perea et al., 2009*; *Santello et al., 2012*).

Interestingly, a number of recent studies have demonstrated a principle of overlap-avoidance among individual astrocytes. Thus, astrocyte processes appear to form no-overlapping domains which may endow each astrocyte with a unique functional coupling to the neighboring neuronal population (*Halassa et al., 2007b*; *Nedergaard et al., 2003*; *Oberheim et al., 2006*). An attractive hypothesis, suggested by the present results, could be that long term and habitual confinement of neuronal activation to specific functional domains may have gradually adapted the individual astrocyte arbor domains to the neuronal functional boundaries in parallel with the dendritic and axonal confinement.

It is important to emphasize that the astrocyte confinement does not necessarily preclude blood flow across functional boundaries- since the capillaries themselves freely cross these boundaries. However, the confinement may allow a more efficient, precise and localized direction of blood flow to functional domains (*Petersen and Sakmann, 2001*). Compatible with this suggestion, optical imaging of blood flow during vibrissae activation reflects a clear confinement of blood supply within the barrel's borders (*Derdikman et al., 2003*; *Woolsey et al., 1996*). Indeed, as has been thoroughly examined in previous studies (*Blinder et al., 2013*), it appears that the highly localized functional selectivity reflected in intrinsic optical imaging is likely due to direct and localized modulation at the level of the microvessels, while more pathological blockage of penetrating arterioles and venules can lead to a more wide-spread vascular effects.

The mechanism that brings about astrocyte structural restriction is currently unknown. If, indeed such confinement serves an adaptive role, an interesting possibility is that this anatomical confinement reflects a dynamic plasticity process. In such a process, the distribution of astrocyte processes may reflect the level of sustained co-activation across neighboring neuronal assemblies, analogous to a Hebbian learning process. Relevant to this hypothesis, previous research has demonstrated activity-dependent dynamic changes in the coupling between astrocyte processes in the case of olfactory glomeruli (*Roux et al., 2011*). It is noteworthy that this hypothesis reverses the causal chain. We do not propose that the morphological shaping of astrocytes somehow endows the underlying tissue with its functional boundaries. Rather, we hypothesize that the astrocyte processes are shaped by the functional selectivity of the underlying neuronal population. Finally, one cannot rule out a more interactive shaping, in which both systems remodel each other.

More generally the finding that astrocyte processes respect functional boundaries could open the way to a variety of plasticity experiments. For example, examining astrocyte remodeling during development (*Diamond et al., 1993*; *Olavarria et al., 1987*), or studying the impact of functional deprivation, such as removing a single or a row of vibrissae on astrocyte morphology. Furthermore, global parameters such as age and pharmacological factors can be examined.

Finally, the generality of the astrocyte confinement requires further examination. It should be noted that previous studies (*Colombo and Reisin, 2004*; *Oberheim et al., 2009*) showed that human astrocytes that populate the superficial cortical layer are larger in diameter (2.6 fold) and show a more complex organization compared to rodents. However, despite these striking differences, our results revealed such confinement in all three systems, including human cortex. It will be interesting to examine whether similar anatomical astrocyte confinements can be observed in other well defined functional boundaries, and the parameters that determine the level of astrocyte confinement in different systems. Our demonstration that astrocyte density modulation can be revealed in micro-structures in human V1 opens the exciting possibility that the non-homogeneity in astrocyte spread could be employed as a new marker for additional areal, columnar and functional discontinuities in the human brain. This is particularly important since the currently available information about such functional boundaries in post-mortem brain tissue is confined to more basic anatomical and histological aspects (*Lorenz et al., 2015*; *Zilles and Amunts, 2010*).

## Materials and methods

### Animals
C57BL/6 mice and Wistar rats were used in this study. The mice and the rats were purchased from Harlan (Jerusalem, Israel). Mice and rats (females), 8–12 weeks of age, were used and kept in a specific pathogen free (SPF) environment. All experiments were approved by the Institutional Animal Care and Use Committee of the Weizmann Institute.

### Human subject
To investigate the striate cortex of the human brain, we examined the right hemisphere of 35-year-old male patient, who died from non-neurological causes. The brain was photographed for purposes of orientation and scale. The occipital lobe through the calcarine sulcus up to the parieto-occipital sulcus, was removed (see *Figure 4—figure supplement 1A*), and was cut into 4 coronal pieces. Two parts were embedded in paraffin blocks while the others were taken for free floating sectioning.

### Preparation of samples for immunohistochemistry
Fluorescein isothiocyanate (FITC)-dextran, (molecular weight 500,000 Daltons, Sigma, St. Louis, MO), 10 mg/ml, was dissolved in PBS and filtered through 0.45 µm mesh. 200 µl of the solution was injected into the tail vein of each animal. Ten min after FITC-dextran injection, mice were deeply anesthetized by a peritoneal injection of ketamin and xylazine (1:1, Kepro, Holland), their brains were removed and post fixed in 2.5% paraformaldehyde in PBS, pH 7.4, for 24 hr.

The left and right cerebral hemispheres of the mice and rats were separated along the longitudinal fissure and the cortex of each side was divided from the underlying white matter. The isolated cortex was then gently flattened between two glass slides, separated by 1.4 mm spacer and soaked for 4–6 days in 1% paraformaldehyde in PBS. The flattened cortices were embedded in paraffin and cut tangentially parallel to the pial surface in thicknesses of 4 or 8 µm by a microtome (Leica, Wetzlar, Germany). Sections were collected and mounted on SuperFrost+ slides (Thermo Scientific, Waltham, MA). Dark field and fluorescent images were taken immediately after the cutting (before the sections were dried) by a Leica M165 FC stereo microscope connected to a digital monochrome camera (Leica, DFC345 FX) at magnifications of X1 and X3.5.

The human primary visual cortex was taken along the banks of the calcarine fissure of the occipital lobe (right hemisphere, age 35). The tissue was embedded in paraffin and cut coronally (from pia to white matter, see *Figure 4—figure supplement 1B*).

### Cytochrome oxidase staining
CO staining followed procedures described previously (*Horton and Hedley-Whyte, 1984*; *Murphy et al., 1995*; *Wong-Riley, 1979*) , with some modifications; sections were incubated in a solution containing 100 mg diaminobmuenzidine, 60 mg cytochrome C (type III, Sigma) and 40 mg catalase per 100 ml of 0.1 M phosphate buffer. The reaction usually took 5d at 39°C. The incubation solution was refreshed daily.

## Immunohistochemistry and microscopy

The staining was done on four neighboring sections per sample, with an interval of about 18–20 μm. Paraffin sections were deparaffinized and rehydrated. Antigen retrieval was performed in 10 mM citric acid pH 6 for 10 min using a low boiling program in the microwave to break protein cross-links and unmask the antigens and epitopes. After pre-incubation with 20% normal horse serum and 0.2% Triton X-100, slides were incubated with rabbit anti-GFAP (1:100, Dako, Glostrup, Denmark) and/or mouse anti non-phosphorylated neurofilament proteins (NPNF, SMI-32; Covance, Dedham, MA, USA).; diluted 1: 100), at 4°C for 7d. To enhance the signal, secondary antibodies, biotinylated anti rabbit or anti mouse (1:100, Jackson ImmunoResearch, West Grove, PA) were added for 90 min, followed by Cy2 or Cy3-conjugated Streptavidin (1:200, Jackson ImmunoResearch West Grove, PA) for 2 hr. Sections were counterstained with Hoechst 33,258 (Molecular Probes, Eugene, OR) for nuclear labeling. For single immunohistochemistry we the *elite*ABC kit (Vector Lab Burlingame, CA, USA) followed by DAB (Sigma) reaction. Stained sections were examined and photographed by a fluorescence or bright field microscope (Eclipse Ni-U; Nikon, Tokyo, Japan) equipped with Plan Fluor objectives (20x; 40x; 60x) connected to a monochrome camera (DS-Qi1, Nikon). To form one large image of the total barrel field area, auditory cortex or V1 cortex, digital images at intermediate magnifications (20x) were collected and stitched together automatically, using NIS element software (Leica). Each large image of the barrel field consists of 60–120 tiles.

In addition, to demonstrate the fine processes of astrocytes reaching the barrel border, some sections were imaged with a confocal scanning microscope (LSM 700, Zeiss). We used an oil immersion objective (100x, NA 1.40; Zeiss). A z-series of the entire 8 μm section was imaged at 0.42 μm spacing followed by a maximum intensity projection.

The staining sensitivity to GFAP was substantially enhanced in the cortex (up to five times) by employing three procedures. First, staining was conducted on thin sections (4–8 μm). Second, an antigen retrieval technique was used, finally we used a long incubation sections with the first antibody (7d at 40°C).

## Alignment of barrel field borders with astrocytes processes and capillaries

Barrel (mice) as well as A1/A2 (rat) boundary demarcation was based on the dark field (contrast inversed) photographs of the sections in layer IV. The boundaries between cortical sublayers IIIa/IIIb were defined based on NPNF staining (*Figure 4A,B*). The borders between blobs/inter-blobs were based on CO staining (*Figure 5D*). These boundaries were then superimposed on the fluorescent images of the astrocytes processes or the capillaries. The superposition was based on careful alignment of the patterns of penetrating arterioles that were identified separately in the two images.

## Image processing and analysis

To reveal the precise details of the distribution of astrocyte processes (GFAP immuno-positive staining) and the capillaries (traced by FITC Dextran) in the barrel field area, manual plotting was performed from the assembled images using Adobe Photoshop software (Adobe Systems, San Jose, CA) at magnifications of 20x. The septal borders were also drawn manually according to the dark field (inversed) images and superimposed on the astrocyte (*Figure 1D*, *Figure 2D and E*) or capillary drawings (*Figure 1B*). The image alignment was done by adjusting the penetrating vessels of the images. A set of lines (375 μm length) was drawn at the centers of the septa, aligned with the border's orientation. For comparison, lines of identical length and orientation were superimposed at the center of the barrel cores. These septa and core related lines were copied and superimposed on the astrocyte and capillary drawings (*Figure 2A–C*). The alignment was done according to the penetrating blood vessels. Only the large barrels were analyzed and only the astrocyte branches/capillaries that crossed the lines were calculated.

To reveal the precise details of the distribution of astrocyte processes (GFAP immuno-positive staining) in A1 *versus* A2 areas, straight lines (50 μm length) were superimposed at a distance of 50 and 200 μm from the border of A1/A2 on both sides aligned with the border's orientation. These A1 and A2 lines were then superimposed on the corresponding drawings of the astrocyte processes (*Figure 3E and G*) and the number of astrocyte branches crossings were calculated.

To quantitatively examine the distribution of astrocytes processes in layer III of the human primary cortex, around the border of IIIa/IIIb, we manually drew the distribution of astrocyte processes at high magnification (see Materials and methods) in both areas. Straight lines (375 μm length) were superimposed on the NPNF image, 50 and 200 μm from the border of the two sublayers, aligned with the border's orientation and then superimposed on the corresponding drawings of astrocyte processes.

To quantitatively examine the relationship between the intensity of cytochrome oxidase and astrocytic GFAP staining along cortical layer II and IIIa we superimposed density plots of both types of staining at three parallel lines and calculated the R-values using Pearson's coefficients of correlation. In addition we examined the distribution of astrocyte processes in blobs/inter-blobs zones. A set of vertical lines (500 μm length) were drawn at the center of blobs, the center of inter-blobs and at the left margin of the inter-blobs. These lines were then copied and superimposed on the astrocyte drawings (*Figure 5D*). The alignment was done according to the pattern of blood vessels. The astrocyte branches that crossed the lines were counted (*Figure 5E*).

All the analysis was performed by a person unaware of the experimental question. For the mice barrel cortex, 8 μm thick, sections were taken one from each of 5 left cortical hemispheres, and on two, 4 μm thick, single sections were taken from two additional hemispheres. The thin sections were analyzed separately to examine possible effects of section thickness on the septa/core crossing ratios. For the rat auditory cortex analysis, we used of one section of each of 4 left hemispheres. For the analysis of the human primary cortex, we used 8 sections from the same brain.

## Statistical analysis

The number of line crossings was compared between septa and barrels (separately for astrocyte processes and capillaries) in the mice barrel cortex, as well as between A1 and A2 in the rat auditory cortex and layer IIIa and IIIb in human primary cortex, using independent-sample Student's t-test. The septa/barrels ratio was compared between astrocyte processes and capillaries using an independent-sample t-test. The septa/barrels ratio was compared between layers (separately for astrocyte processes and capillaries) using a one-way ANOVA on log-transformed ratios, followed by a Tukey post-hoc test where appropriate. All tests were performed by Statsoft's Statistica, version 12.

## Acknowledgement

The authors thank Menahem Segal and Haidarliu Sebatian for helpful advice and comment, Thanks are due to Yuri Kuznetsov and Dor Gravits for technical help and data collection assistance, Calanit Raanan and Marina Cohen for help in the preparation of histological samples, Haya Avital for the graphic work, and Ron Rotkopf for the statistical analysis.

## Additional information

### Funding

| Funder | Grant reference number | Author |
|---|---|---|
| Weizmann Institute of Science | Krenter Institute - Equipment grant support | Raya Eilam |
| European Commission | EU EP7 VERE | Rafael Malach |
| Israeli Centers for Research Excellence | Israel Science Foundation grant No.51/11 | Rafael Malach |

The funders had no role in study design, data collection and interpretation, or the decision to submit the work for publication.

### Author contributions

RE, Conceived the project, Performed the experiments, Analyzed the data, Wrote the paper; RAh, Performed the experiments, Conception and design, Drafting or revising the article; RAr, Conception and design, Drafting or revising the article; RM, Conceived the project, Analyzed the data, Wrote the paper, Acquisition of data

## Author ORCIDs

Rafael Malach, http://orcid.org/0000-0002-2869-680X

## Ethics

Human subjects: Right human brain occipital lobe was provided by the Netherlands Brain Bank
Animal experimentation: All the animal experiments were approved by the Institutional Animal Care
and Use Committee of the Weizmann Institute. Protocol number 20640915-2.

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
