## [Decision Letter]

Thank you for submitting your article "Astrocyte Morphology is Confined by Cortical Functional Boundaries in Mammals Ranging from Mice to Human" for consideration by *eLife*. Your article has been reviewed by three peer reviewers, including Christian Giaume and a member of our Board of Reviewing Editors, and the evaluation has been overseen by Sabine Kastner as the Senior Editor.

The reviewers have discussed the reviews with one another and the Reviewing Editor has drafted this decision to help you prepare a revised submission.

Summary:

This manuscript reports a morphometric analysis of astrocytes in cortical brain regions from mouse and human where functional boundaries are known to occur. This work is well conducted and data are strongly convincing.

Essential revisions:

1) Given the straightforward nature of the study the manuscript is laborious. The story could easily be told in fewer pages and with fewer figures. For example, all of the supplementary figures are impressive but essentially repeat what is shown in Figure 1 to 5. Please condense and remove supplementary figures.

2) The story is told from an angle that appears infused by the assumption that the parcel-specific distribution of astrocytes is shaped by neuronal activity within a functional area. This might be so, but the study provides no evidence for the idea other than that the density of astrocyte processes varies with parcel-borders. Although stated at the outset that anatomical remodeling" occurs over time there is no experimental investigation of such a process. Thus, please eliminate all of the suggestive terminology that refers to: "shaped", sculpting, remodeling, and reshaping. All of these terms are interpretations and are better suited for the Discussion section.

3) The authors consider the astrocytes distribution and shape only in term of blood flow control, however, it is now well established that the processes of these glial cells also establish close contacts with synapse and form the so called "tri-partite synapse". Consequently, these specific features of astrocytes described here should also be discussed in term of neuroglial interaction.

4) In addition, it is now well established that astrocytes occupy domains that do not overlap. Consequently, the authors should discuss their results also in reference to these particularities that are not shared with neurons. There is a real need to present the data in reference to the literature.

---

## [Author Response]

*Essential revisions:*

*1) Given the straightforward nature of the study the manuscript is laborious. The story could easily be told in fewer pages and with fewer figures. For example, all of the supplementary figures are impressive but essentially repeat what is shown in Figure 1 to 5. Please condense and remove supplementary figures.*

Following the reviewer comment we have now removed all the supplementary figures except Figure 4—figure supplement 1 which we think is important in providing a needed perspective on the scales involved in human brain anatomy.

*2) The story is told from an angle that appears infused by the assumption that the parcel-specific distribution of astrocytes is shaped by neuronal activity within a functional area. This might be so, but the study provides no evidence for the idea other than that the density of astrocyte processes varies with parcel-borders. Although stated at the outset that anatomical remodeling" occurs over time there is no experimental investigation of such a process. Thus, please eliminate all of the suggestive terminology that refers to: "shaped", sculpting, remodeling, and reshaping. All of these terms are interpretations and are better suited for the Discussion section.*

We thank the reviewer for noting this potential ambiguity in our terminology. Indeed the present data does not resolve the mechanism that brings about the astrocyte confinement. Following the reviewer's suggestion we have now replaced all the ambiguous terms with the more anatomically descriptive terms "confinement" or "restriction".

*3) The authors consider the astrocytes distribution and shape only in term of blood flow control, however, it is now well established that the processes of these glial cells also establish close contacts with synapse and form the so called "tri-partite synapse". Consequently, these specific features of astrocytes described here should also be discussed in term of neuroglial interaction.*

We agree with the reviewer that neuro-vascular coupling is not the exclusive source of neuron-astrocyte interaction. We have now added the following paragraph in the Introduction:

"Another interesting aspect of astrocyte morphology is the tight coupling between the neuronal and astrocyte processes. In particular, recent evidence has demonstrated the existence of tight anatomical co-localization of astrocyte processes and neuronal synapses in what has been termed a "tri-partite synapse" formation (Araque et al., 1999; Halassa et al., 2007a; Perea et al., 2009; Santello et al., 2012). Such links further suggest that astrocyte morphology may follow boundaries defined by neuronal functional compartments."

And in the Discussion:

"Furthermore, astrocyte processes have been demonstrated to manifest tight anatomical formations with neuronal dendrites and the activity of the brain actually arises from the coordinated activity of a network comprises of both neurons and astrocytes (Araque et al., 1999; Halassa et al., 2007a; Perea et al., 2009; Santello et al., 2012)."

4) In addition, it is now well established that astrocytes occupy domains that do not overlap. Consequently, the authors should discuss their results also in reference to these particularities that are not shared with neurons. There is a real need to present the data in reference to the literature.

Following the reviewer's comment we have now added the following paragraph in the Discussion:

"Interestingly, a number of recent studies have demonstrated a principle of overlap-avoidance among individual astrocytes. Thus, astrocyte processes appear to form no-overlapping domains which may endow each astrocyte with a unique functional coupling to the neighboring neuronal population (Halassa et al., 2007b; Nedergaard et al., 2003; Oberheim et al., 2006). An attractive hypothesis, suggested by the present results, could be that long term and habitual confinement of neuronal activation to specific functional domains may have gradually adapted the individual astrocyte arbor domains to the neuronal functional boundaries in parallel with the dendritic and axonal confinement."